# Cross-Talk between Overlap Interactions in Biomolecules: A Case Study of the *β*-Turn Motif

**DOI:** 10.3390/molecules26061533

**Published:** 2021-03-11

**Authors:** Jayashree Nagesh

**Affiliations:** Solid State and Structural Chemistry Unit, Indian Institute of Science Bangalore, Bengaluru 560012, Karnataka, India; jayashreen@iisc.ac.in

**Keywords:** beta-turns, noncovalent interactions, n-pi*, protein secondary structure, protein conformation, protein stability, NBO deletion analysis, bioinformatics

## Abstract

Noncovalent interactions play a pivotal role in regulating protein conformation, stability and dynamics. Among the quantum mechanical (QM) overlap-based noncovalent interactions, $n\to \pi^{*}$ is the best understood with studies ranging from small molecules to β-turns of model proteins such as GB1. However, these investigations do not explore the interplay between multiple overlap interactions in contributing to local structure and stability. In this work, we identify and characterize all noncovalent overlap interactions in the β-turn, an important secondary structural element that facilitates the folding of a polypeptide chain. Invoking a QM framework of natural bond orbitals, we demonstrate the role of several additional interactions such as $n\to \sigma^{*}$ and $\pi \to \pi^{*}$ that are energetically comparable to or larger than $n\to \pi^{*}$. We find that these interactions are sensitive to changes in the side chain of the residues in the β-turn of GB1, suggesting that the $n\to \pi^{*}$ may not be the only component in dictating β-turn conformation and stability. Furthermore, a database search of $n\to \sigma^{*}$ and $\pi \to \pi^{*}$ in the PDB reveals that they are prevalent in most proteins and have significant interaction energies (∼1 kcal/mol). This indicates that all overlap interactions must be taken into account to obtain a comprehensive picture of their contributions to protein structure and energetics. Lastly, based on the extent of QM overlaps and interaction energies, we propose geometric criteria using which these additional interactions can be efficiently tracked in broad database searches.

## 1. Introduction

Noncovalent interactions (NCIs) have been a central topic of interest in chemistry ever since the conception of chemical bonding ideas [1]. There is a recent surge of interest in the role of such interactions in biology with increase in the ability to handle large amounts of data involving molecular structures as well as advancements in the identification and classification of different interactions. The last two decades have brought forth the importance of NCIs such as π-π, ion-π, and lone-pair-π, to add to the existing electrostatic, hydrogen bonding and hydrophobic effects in protein stability [2,3], enzyme catalysis [4,5], molecular recognition [6,7] and phase separation [8]. These interactions are gaining importance not only as factors in modulating structure and stability, but also in facilitating kinetic events in molecular recognition and enzyme catalysis.

The importance of quantum mechanical (QM) orbital overlap based interactions in small molecules (<100 atoms) is well-established. Such interactions are typically detected based on the contact distance between a pair of atoms being smaller than the sum of their van der Waals radii [9], and could be short-range filled-filled repulsive type or long-range attractive donor-acceptor type. Several recent reports have shown that donor-acceptor groups can participate in more than one type of interaction depending on their relative orientations. For instance, the C=O group is known to engage in hydrogen bonding (C=O⋯H) [10], carbonyl-chalcogen (C=O⋯X where X can be Sulphur, Selenium or Tellurium) [11], and nucleophile-carbonyl (Burgi-Dunitz trajectory type) interactions [12,13]. The last group of aforementioned interactions, involving $n\to \pi^{*}$ and $\pi \to \pi^{*}$ overlaps [14] have been found in dimers of acetone and formaldehyde. Additionally, the C=O group orbitals can also overlap with σ orbitals [14] in its proximity. Other examples of the co-existence of several long-range interactions include $n\to \pi^{*}$ and hydrogen bonds [15,16], CO-CO and hydrogen bonds [17], and cation-π, anion-π and π-π [18]. Hence it is clear that the structure and electronic stability of a molecule is intimately governed by the interplay of multiple overlap interactions in addition to classical interactions such as salt bridges.

In comparison to small molecules, there are far fewer investigations of overlap interactions in biomolecules, and studies have primarily focused on classical electrostatic and hydrogen bonding interactions [19,20]. Some of the earliest overlap-based studies were centered around using the natural bond orbital (NBO) approach to understand hydrogen bond networks (see for example [21,22]) that are known to stabilise the native state of proteins [23,24]. Over the years, the characterisation of multiple (for example in hydrogen bond networks in DNA) [21,22] and multi-valent (for example in π-π networks in liquid-liquid phase separation) [8] interactions have revealed the complexity in understanding the basis of biomolecular structure.

The $n\to \pi^{*}$ interaction is the next best studied overlap type of interaction in biomolecules after hydrogen bonding [25,26]. It involves the sharing of electrons between a filled non-bonding orbital of a heteratom (like O, N, S) and an empty $\pi^{*}$ orbital nearby. In β-turns of proteins, this is an inter-residue interaction comprising of $O_{i}$(n)→(C=O)${}_{i+1}$($\pi^{*}$) or $O_{i+1}$(n)→(C=O)${}_{i+2}$($\pi^{*}$). Recent experiments have established the presence of this interaction spectroscopically [27,28,29] in small molecules, and through melting temperature and free energy measurements in proteins [30]. In the context of proteins, Khatri et al. [30] examined the influence of $n\to \pi^{*}$ interaction on protein stability by engineering the sequence of a β-turn, thereby affecting the strength of the interaction itself. It was seen that changing the side chain identity of the residues in the β-turn resulted in a good correlation between the computationally estimated $n\to \pi^{*}$ interaction energy and the melting temperature of the proteins, suggesting that one could control protein stability by systematically altering the $n\to \pi^{*}$ interaction of the β-turn. Although the $n\to \pi^{*}$-melting temperature correlation is encouraging, the role of local electronic factors, such as other interactions, the solvent and the unfolded state remains unclear.

Here we identify additional QM overlap based interactions such as $n\to \sigma_{CH}^{*}$, $n\to \sigma_{CC}^{*}$ and $\pi \to \pi^{*}$ in a β-turn of the model protein GB1 (PDB:2QMT). We find that upon side chain substitution, these interactions show compensatory effects in terms of their electronic stabilization of the local β-turn conformation. We explore the prevalence of these overlap interactions in β-turns of several protein structures in the protein data bank (PDB) and find that they are prominent with interaction strengths of ∼1 kcal/mol. Taken together, our results suggest that a holistic view of all interactions is necessary to understand the influence of NCIs on biomolecular structure and function.

## 2. Results

### 2.1. Identifying Overlap-Based Interactions in β-Turns of Proteins

The β-turn is an important secondary structure motif that permits a change in the direction of a peptide chain. A β-turn is characterised by a chain of four amino acids, labeled as *i*, i+1, i+2 and i+3 (Figure 1). It is an ideal model system for studying key interactions responsible for directing and stabilising the folding of the peptide chain. Recently, Khatri et al. [30] examined the effect of structural changes in the loop L1 β-turn of a model protein GB1 (Appendix A) on the strength of the $n\to \pi^{*}$ interaction and in turn on protein stability. Four β-turn variants of GB1 were synthesised by substituting Lys10 at the i+1 position of L1 by D-valine(“v”) and Thr11 at the i+2 position by the L-amino acids alanine, serine, valine and threonine to obtain “vA”, “vS”, “vV” and “vT” variants respectively. The use of D-valine fixes the type of the β-turn to be II’, and systematic engineering of the i+2 position creates differences in the backbone angles $\phi (C-N-C_{\alpha }-C)$ and $\psi (N-C_{\alpha }-C-N)$, that in turn tunes the i+1–i+2 inter-residue $n\to \pi^{*}$ interaction. Using the NBO approach on the crystal structures of these β-turn variants (6L9B (vA), 6L91 (vT), 6L9D (vS) and 6LJI (vV); also see SI of Ref. [30]) it was found that the energy of the i+1–i+2$n\to \pi^{*}$ interaction in these β-turns is in the range 0.20–0.46 kcal/mol. Here, we first explore the presence of other long-range interactions in β-turns that may be comparable in energy to $n\to \pi^{*}$ and hence likely play as much of an important role as the $n\to \pi^{*}$. In order to make systematic comparisons with previous studies, we initially focus on the i+1–i+2 residue pair.

The process of identifying noncovalent interactions is shown in Figure 1B. All calculations in this paper were carried out with wB97XD [31]/6-31+G(d,p) model chemistry using the Gaussian suite of programs [32] in conjunction with the NBO 6.0 package [33]. The tetrapeptide alone was used in the quantum chemical calculations with the *i* and i+3 residues capped off with hydrogen atoms (see SI for coordinates). Only the positions of the hydrogen atoms were optimized, and the rest of the atomic positions were kept the same as the crystal structure. A polarizable continuum model [34] (Section 4.2) was used to implicitly account for the water solvent. To identify inter-residue interactions, (a) overlaps within a peptide bond or those involving a common atom were excluded, and (b) overlaps involving atoms or groups separated by at least two covalent bonds were considered; this ensures that we do not include overlaps due to rotation about a covalent bond. The criterion of $\Delta E^{\left(2\right)}>0.1$ kcal/mol (see Section 4) was chosen to screen for the interactions that are of comparable or higher strength than the $n\to \pi^{*}$. This exhaustive procedure of sifting through all possible overlaps and interaction energies followed by specific screening criteria revealed numerous and previously neglected interactions apart from the $n\to \pi^{*}$ (Figure 2 and Table 1).

### 2.2. Analysis of NCIs in β-Turns in Terms of Structure and Interaction Energy

Once all the spatially overlapping interactions were identified, we subjected them to a NBO-deletion analysis (see Section 4) to estimate their stabilising effect on the electronic energy of a given geometry of the tetrapeptide. The deletion energy is a useful indicator for gauging the effect of a particular interaction to the electronic energy since it is based on the extent of overlap between the donor and acceptor orbitals in this interaction (see Section 4). Figure 3 shows a comparison amongst the energetic contributions of all the four overlap interactions in each of the β-turn variants. Upon change of side chain identity, there are changes to the individual interaction energies, but the overall energy stays about the same (1–1.2 kcal/mol). This is an intriguing observation because although a mutation is expected to bring about local changes in the electronic interactions, the total interaction energy locally seems to maintained approximately the same. This constancy of the overall energy comes from a compensatory effect of the interactions. In particular, it is known from previous studies [30] that the vV variant does not engage in an $n\to \pi^{*}$ interaction. The identification of additional interactions reveals that vV is instead stabilised by $n\to \sigma_{CH}^{*}$ and $n\to \sigma_{CC}^{*}$. The $\pi \to \pi^{*}$ interaction mirrors the trends shown by $n\to \pi^{*}$ as expected, since both involve interaction between the carbonyl groups of adjacent residues. Upon variation of the side chain, we see that the $n\to \sigma_{CC}^{*}$ is the least sensitive in its strength, indicating that it might not be important in relative stabilization studies of these β-turns. However this lack of sensitivity may not be a universal trend, and remains to be confirmed by examination of β-turns in other proteins (see below).

In order to understand the above results, we examine the structural variations induced by the side chain substitution of the loop L1 variants of GB1 protein. Figure 4 and Table 2 show how various internal coordinates, including the backbone angles of ϕ and ψ are affected by changing the i+2 residue. The internal coordinates identified here come from simple considerations. In the $n\to \sigma_{CC}^{*}$ interaction, the $O^{i+1}$, $C_{\alpha }^{i+2}$ and $C_{\beta }^{i+2}$ are involved, hence the angle τ is a prudent choice. Similarly, in $n\to \sigma_{CH}^{*}$ and $\pi \to \pi^{*}$ interactions, the dihedral angles ξ and Δ are obvious choices. However, these dihedral angles are not sufficient to characterise these interactions since they cannot describe the lateral overlap of the orbitals. Hence we also require the use of distance criteria Ω and λ to quantify the interaction strengths.

Upon changing the i+2 residue from threonine to valine, its (ϕ,ψ) values increasingly deviate from the α-helical values of $\left(-60^{\circ },-45^{\circ }\right)$ and accordingly cause the $n\to \pi^{*}$ interaction to decrease in its strength [30]. This trend is mirrored by the $\pi \to \pi^{*}$ interaction because of the increase in λ (O${}^{i+1}$–O${}^{i+2}$ distance). In the case of vV, although Δ is similar to that in vA, there is no $\pi \to \pi^{*}$ interaction because of the large λ. This suggests that for $\pi \to \pi^{*}$ interaction, λ has to be small enough before considering the Δ coordinate to rationalize the extent of overlap.

For understanding the trends in the two $n\to \sigma^{*}$ interactions we analysed the Ω, ξ and τ coordinates. In a comparison of the four β-turn variants, the τ values are about the same, reflecting the near constancy of the $n\to \sigma_{CC}^{*}$ interaction strengths. We will see that this is not necessarily the case in the β-turns of other proteins. The $n\to \sigma_{CH}^{*}$ on the other hand represents a contrasting situation: as the $n\to \pi^{*}$ becomes unfavourable from threonine to valine (with decreasing helix propensity), the $n\to \sigma_{CH}^{*}$ becomes favourable due to the decrease in ξ angle with increase in ψ. The decrease in ξ here is also accompanied by the decrease in the Ω (O${}^{i+1}$–$H_{\alpha }^{i+2}$ distance) suggesting that in this peptide series, $n\to \sigma_{CH}^{*}$ plays a compensatory role in locally stabilizing the turn when $n\to \pi^{*}$ becomes unfavourable. These results for the four variants thus help in obtaining an atomic level picture of how the microscopic interplay of interactions could help maintain an overall local electronic stability of ∼1 kcal/mol in the engineered β-turn of GB1.

### 2.3. Prevalence of NCIs in β-Turns of PDB Structures

#### 2.3.1. Energetics

In order to test for the prevalence of the numerous overlap interactions highlighted in this work, we chose 22 proteins from the set examined by Khatri et al. [30] (see Appendix A for the list of proteins with one selected β-turn per protein and the relevant internal coordinate values with energies and Appendix A for the amino acid sequence of the β-turn) and subjected them to same kind of analysis (Figure 1B) as the GB1 β-variants to characterize the four overlap interactions. The list of 22 proteins includes all the four types of β-turns (I,I’,II,II’) with randomly chosen structures in each type. Figure 5A shows the total stabilization energy due to these interactions, and it is evident that in contrast to the GB1 variants, NCIs in these proteins contribute varying extents to the electronic energy of the protein. This is primarily because the i+1 residue is fixed in GB1 variants whereas here both i+1 and i+2 residues vary from one protein to another. Another interesting observation is that by no means is the $n\to \pi^{*}$ interaction the dominant interaction in cases where it is present; we see instances where the $n\to \sigma_{CH}^{*}$ electronic energy stabilization is equal to or even greater than $n\to \pi^{*}$. This analysis strongly supports the notion that several interactions comparable to $n\to \pi^{*}$ may play a role in influencing protein structure and stability.

#### 2.3.2. Geometric Criteria Based on Variations in β-Turn Sidechain Structure

The internal coordinates defined in Figure 4 are tabulated for the β-turns in the 22 proteins chosen above (Appendix A). We observe that some of the trends in the GB1 variants are corroborated here as well, though not in all the cases. An examination of the geometric values reveals the following parameter ranges within which there are non-negligible overlaps leading to stabilisation.

As λ increases, $\pi \to \pi^{*}$ decreases in strength. A small value of this coordinate is more important than the dihedral Δ in order for the $\pi \to \pi^{*}$ interaction to materialise. From this analysis, it is clear that this interaction exists when 0 Å<λ<3.7 Å and $0^{\circ }<\Delta <28^{\circ }$.The strength of the $n\to \sigma_{CC}^{*}$ interaction decreases as τ deviates from $180^{\circ }$, suggesting that this interaction is likely to be present when $130^{\circ }<\tau <180^{\circ }$. Though the lower limit of $130^{\circ }$ has emerged from the current analysis of only 22 proteins, it is apparent that the extent of overlap will go to zero as we approach $\tau =90^{\circ }$ or lower.The $n\to \sigma_{CH}^{*}$ presents two possible scenarios for its existence. The $\sigma^{*}$ orbital can overlap with the *n* orbital either when C=O and $C_{\alpha }-H_{\alpha }$ are on the same side of the peptide backbone, or on the opposite sides of the backbone because of the cylindrical nature of the $\sigma^{*}$ orbital. Therefore this interaction is possible when (a) 2.2 Å<Ω<2.6 Å and $|\xi |<53^{\circ }$, and (b) 3.8 Å<Ω<4.0 Å and $160^{\circ }<\xi <200^{\circ }$. Though the latter case was not seen in the GB1 variants, the PDB analysis shows that these interactions can occur in more than one orientation. Note that this is in contrast to expecting this interaction to exist simply based on the distance Ω being less than sum of van der Waals radii of $O^{i+1}$ and $H_{\alpha }^{i+2}$, and an exhaustive NBO analysis was essential to establish the criterion for the existence of $n\to \sigma_{CH}^{*}$ even when Ω is large.

#### 2.3.3. (ϕ,ψ) Analysis

In order to place the four overlap interactions within the context of protein backbone dihedral angles, we performed a Ramachandran map analysis (Figure 5B) where we depict each of the 22 proteins by a point on this map, since there is only one β-turn chosen per protein. A given β-turn can engage in multiple interactions, and this is depicted by the contributions in Figure 5A. For example, the point at ((105${}^{\circ }$,$-14^{\circ }$) of i+2 residue in the protein with PDB ID 1C0I in the residue range 1186–1189) has both $n\to \pi^{*}$ and $n\to \sigma_{CH}^{*}$ interactions. The $\pi \to \pi^{*}$ is the least prevalent interaction in terms of the number of β-turns it occurs in (see Appendix A for the data used to generate this plot) and is also the weakest among the interactions. The Ramachandran map plot above also indicates that if the $\pi \to \pi^{*}$ interaction is present, there should be an $n\to \pi^{*}$ interaction as well, though the reverse is not necessarily true because of the requirement of lateral overlap of the π and $\pi^{*}$ (see Figure 2) orbitals. In contrast to the lack of variation in $n\to \sigma_{CC}^{*}$ interaction in the β-turn variants in GB1 above, the PDB analysis reveals at least 0.5 kcal/mol variation between different β-turns. This suggests that this interaction might also play a tangible role in the local stability depending on the structural context in which it occurs. Turning to the $n\to \sigma_{CH}^{*}$ interaction, we see the widest variation in strength ranging from 0–2.1 kcal/mol, and its presence in turns where no other long-range interactions are evident (for instance proteins 8 and 9). These two observations along with the contributions shown in Figure 5A strongly suggest that a comprehensive view of all interactions is essential in understanding the role of NCI in protein conformational changes.

## 3. Discussion

NCIs form an integral part of the molecular basis determining biomolecular structure, kinetics and stability. In this paper we have identified several QM overlap based interactions in the β-turn motif in the GB1 model protein, and have found that in addition to the $n\to \pi^{*}$ interaction, $n\to \sigma_{CC}^{*}$, $n\to \sigma_{CH}^{*}$ and $\pi \to \pi^{*}$ play an important role in dictating the electronic stability of the β-turn tetrapeptide. A PDB analysis of these interactions also reveals the presence of all these interactions in β-turns and indicates that $n\to \pi^{*}$ is not the dominant long-range overlap interaction in several cases.

The aforementioned procedure of identifying donor-acceptor interactions using NBO analysis is general and can be applied to any structural motif in biology. The notion of interaction energy is generally defined in the context of two molecules that are not covalently bonded; however, the NBO approach can aid in the classification of interactions within the same molecule [36] (see Section 4). An important consideration in using this protocol is the size of the motif that is tractable using quantum chemistry methods. A system containing about 50 atoms or less can be studied rigourously and efficiently using the approach in this paper. Once the different types of interactions in various motifs are identified, one can use more efficient methods of tracking interactions in the biomolecule based on robustly defined geometric criteria.

Our aim is primarily to investigate the complimentary and supplementary roles of attractive overlap interactions in β-turns. In addition to donor-acceptor type attractive interactions, there are filled-filled interactions between orbitals that are repulsive in nature [37]. These filled-filled interactions therefore can be antagonistic to attractive donor-acceptor interactions [38] in terms of defining steric boundaries between various groups. However, these interactions are likely to become important only for short contacts. An integrated view of the electronic energetics contributing to peptide stability therefore will require a consideration of such filled-filled interactions. This is beyond the scope of this study because we are not looking to make an exhaustive list of contributions to the peptide enthalpy and are only demonstrating the need to acknowledge the presence of multiple attractive interactions of similar energy.

It is well-established that the hydrophobic effect contributes to the bulk of the free energy change in the stability of a protein [39,40]. The NCIs, including hydrogen bonds, contribute far less (<5%) towards stability [23,40]. A single NCI is weak (<1 kcal/mol) and not strong enough to compete with the forces of the hydrophobic effect. However, there are studies suggesting that the cooperativity amongst the NCIs [41] is capable of tuning the specific folding pattern and functioning of RNA [42], DNA [43,44,45], carbohydrates [7] and proteins that mediate liquid-liquid phase separation [8,46,47]. Likewise, we suggest that even in a small motif such as the β-turn, it is the cooperativity of various interactions that dictates the formation of specific conformations. Since the interactions may complement or supplement each other, they therefore must be considered together.

The measurement of melting temperatures for each of the proteins containing the β-turn variants in Ref. [30] showed that the decreasing trend in $n\to \pi^{*}$ interactions correlated with a decrease in the melting temperature. In order to describe the molecular basis of the decrease in the melting temperatures it will be important to consider the effect of side chain substitution at i+2 on both the unfolded and folded states, and in particular the role of solvent in differential stabilization of the unfolded state, if any [23]. We anticipate that if indeed the reduction in $n\to \pi^{*}$ strength is the cause of the decrease in the melting temperature, this will be substantiated by the canceling effects of the long-range intra- and inter-molecular interactions in the unfolded and folded states. This will be the subject of an upcoming study.

## 4. Methods

### 4.1. Natural Bond Order Analysis

The natural bond order analysis [48] is one of the several [49,50,51,52,53,54] ways of analysing electronic wavefunctions and densities generated by an *ab initio* quantum chemistry package. The NBO approach is one of the most consistent methods to rationalise chemical behaviour in relation to established knowledge, and has a predictive utility for new or unexpected directions. The NBOs are a result of a series of transformations from the atomic orbitals (AOs) of a chosen electronic basis through the formation of “natural AOs” [48] based on the Carlson-Keller theorem [55]. Each NBO calculation results in a set of ‘Lewis-type’ (donor) and non-Lewis-type (acceptor) orbitals. Unlike molecular orbitals where there is a clear demarcation between occupied and vacant orbitals, the donor and acceptor NBOs have partial occupancies of electrons. The partial occupancy of the acceptor orbitals represent the ‘delocalization’ (or deviation) of the donor orbitals from an ‘idealised’ Lewis-type localised representation. Thus the acceptor orbitals lead to lowering of the covalent bond energy and change in the overall wavefunction. In the perturbative limit where the noncovalent or delocalised contributions comprise <1% of the covalent contribution, the energy of such donor-acceptor type of interaction is estimated as
(1)$$\begin{array}{cc} \Delta E_{DA}^{\left(2\right)} & =-2\frac{\langle \Phi_{D}|\hat{F}{|\Phi_{A}\rangle }^{2}}{\epsilon_{A}-\epsilon_{D}} \end{array}$$
where $\hat{F}$ is the Fock operator [56], $\epsilon_{D}$ and $\epsilon_{A}$ are NBO energies, and {$\Phi_{D}$ and $\Phi_{A}$} are the NBOs. This expression assumes minor energy changes to a selected donor orbital $\phi_{D}$ due to the ‘interaction’ of this orbital with the $\phi_{A}$. However, interactions of $\phi_{A}$ with other donor orbitals can take place, and higher order effects need to be taken into account to get the actual change in the energy of the system due to this donor-acceptor pair. This is achieved using a deletion approach, where the interaction terms $\langle \Phi_{D}|\hat{F}|\Phi_{A}\rangle$ and $\langle \Phi_{A}|\hat{F}|\Phi_{D}\rangle$ are simply deleted from the full Fock matrix. The Fock matrix is then rediagonalised to get new NBOs, NBO energies and the overall ground electronic state energy (${\tilde{E}}_{0}$). The resulting NBO energy for this donor orbital will be higher, and we thus get the true stabilisation energy $\Delta E_{DA}=E_{0}-{\tilde{E}}_{0}$ of the molecule due to the presence of this donor-acceptor interaction. $E_{0}$ and ${\tilde{E}}_{0}$ refer to the ground electronic state energy before and after the deletion of the Fock matrix elements respectively.

### 4.2. Solvent Model

The polarizable continuum solvent (PCM) model involves treating the solvent as a continuum dielectric medium that interacts only electrostatically with the solute [34]. The solute is placed in void cavity that contains within its boundaries the largest possible part of the solute charge distribution. The shape of the cavity broadly outlines the molecular shape in order to avoid any unrealistic deformation due to solvent polarization. The charge distribution of the solute inside the cavity polarizes the dielectric continuum which in turn polarizes the solute charge distribution. This interaction is defined in a self-consistent manner that is numerically solved using an iterative procedure. The implicit solvent models in literature differ in the size and shape of the solute cavity, level of solute description, dielectric medium description, cavity calculation method and representation of charge distribution. In the default model used by the Gaussian program (Integral Equation Formalism-PCM, or IEFPCM) a spherical cavity is used. The solute is described using the model chemistry wB97XD/6-31+G(d,p). The dielectric medium has a fixed dielectric constant (ϵ), and the reaction field is modeled using a continuous surface charge formalism that ensures its smoothness [57]. The mathematical problem involves the solution of the general Poisson equation for the electrostatic potential (V) involving the molecular charge density $\rho_{mol}$
(2)−∇2V=4πρmolinsidethecavity,(3)−ϵ∇2V=0outsidethecavity,(4)V→0(∞)where(5)V=Vmol+VR
accompanied by conditions [34] to ensure continuity of the potential across the cavity surface, and $V_{R}$ is the reaction potential described by an apparent surface charge density on the cavity using Green’s functions in the IEF-PCM approach. A detailed description of these functions is outside the scope of this paper, and the interested reader is referred to the review by Tomasi et al. [34].

## Figures and Tables

**Figure 1 molecules-26-01533-f001:**
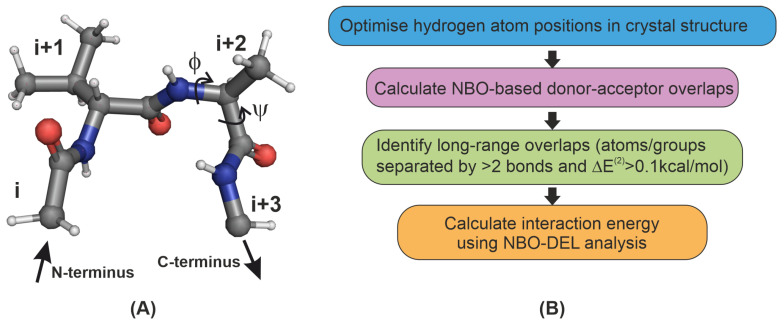
(**A**) Typical tetrapeptide showing the i+1 and i+2 residues at a β-turn and the backbone (ϕ,ψ) angles used in this work. (**B**) Flowchart of the steps used in identifying donor-acceptor interactions. See Section 4 for a description of the second order perturbation energy $\Delta E^{\left(2\right)}$.

**Figure 2 molecules-26-01533-f002:**
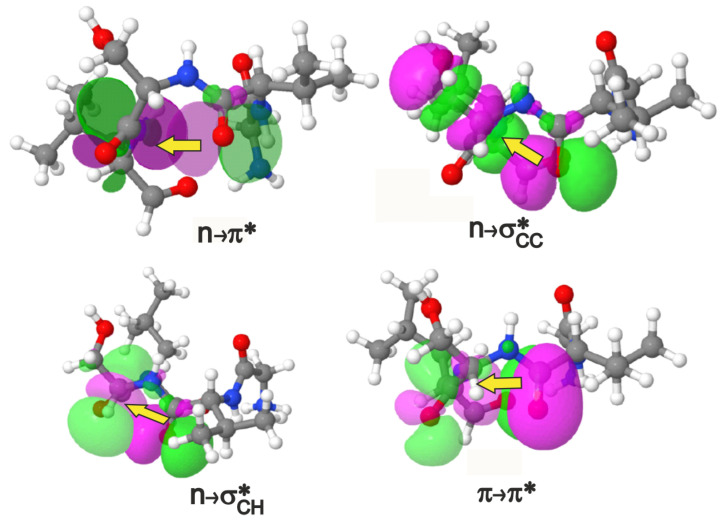
The $n\to \pi^{*}$, $n\to \sigma^{*}$ (2 types) and $\pi \to \pi^{*}$ interactions shown in terms of donor-acceptor orbital overlaps in a view where the i+1–i+2 residues project out of the plane of the paper towards the reader. An isovalue of 0.02 e/Å${}^{3}$ was used in the plots. The i+1–i+2 pairs shown above are valine-serine for the depiction of $n\to \pi^{*}$, $n\to \sigma_{CH}^{*}$ and $\pi \to \pi^{*}$, and valine-threonine for $n\to \sigma_{CC}^{*}$. The magenta/green stand for opposite phases of the orbital wavefunction. If we choose to assign positive values to the magenta lobe of the orbital in that region of 3D space, then the green lobe represents negative values of the same orbital and vice versa.

**Figure 3 molecules-26-01533-f003:**
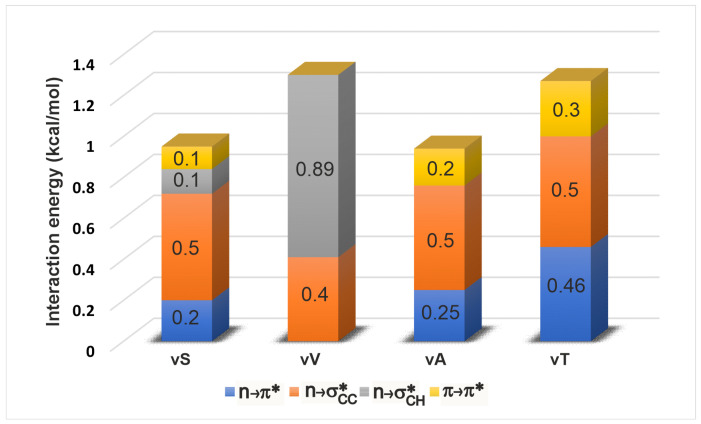
Energies of comparable long-range interactions obtained using NBO deletion analysis. It is seen that upon change in the amino acid (and therefore the side chain) in the i+2 position, each type of interaction increases or decreases to maintain an overall contribution of ≈1 kcal/mol to the lowering of peptide electronic energy.

**Figure 4 molecules-26-01533-f004:**
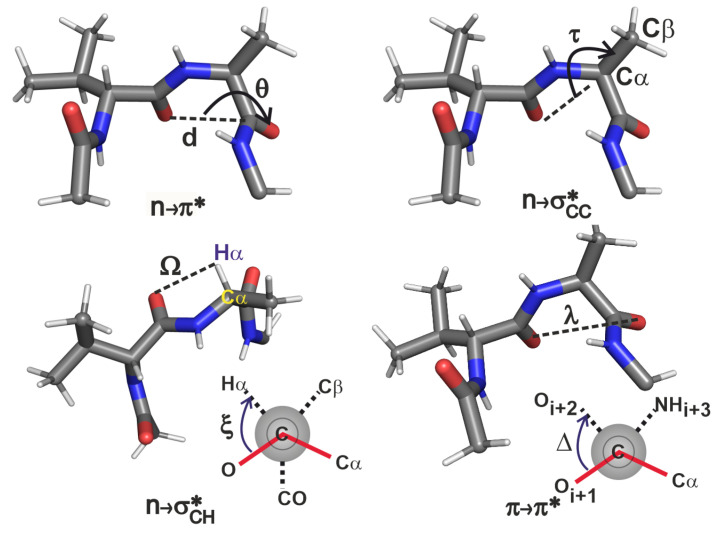
Internal coordinate definitions for analysing each interaction; the peptide shown for each interaction is only illustrative; all the four interactions are analysed for each peptide. The ξ dihedral angle is shown as a Newmann projection, where the C=O of the i+1 residue is present in front of the plane of the paper, the $C_{\alpha }^{i+2}$ is drawn as a grey circle behind the plane of the paper, and the NH of i+2 residue is not shown. The Δ dihedral angle is also shown in the Newman projection where the C=O of the i+1 and i+2 residues are in the front and back of the plane of the paper respectively, and the $C_{\alpha }$ and NH groups of the i+2 residue are not shown.

**Figure 5 molecules-26-01533-f005:**
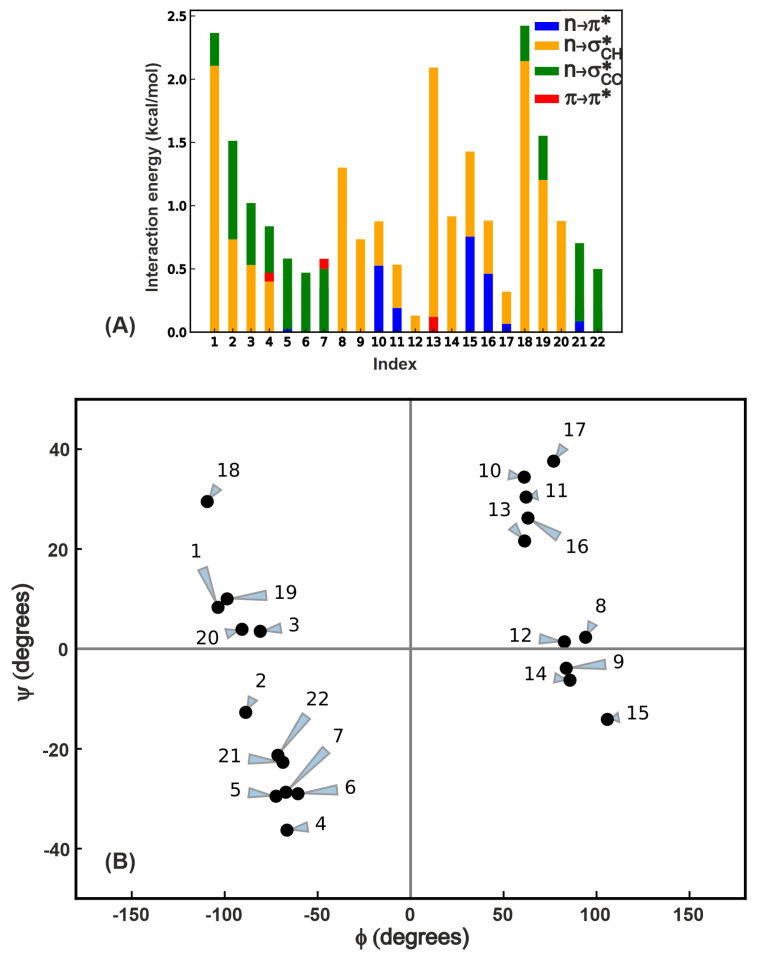
(**A**) Total energy of all long-range interactions in a subset of proteins from the PDB [35] that have β-turns. (**B**) Ramachandran plot of i+2 residues engaged in four types of i+1–i+2 interactions. Each point indicates the (ϕ,ψ) of an i+2 residue of a β-turn in one of the 22 PDB proteins. There may be more than one interaction for a given i+2 residue, and this is indicated by the bar plot in (**A**). See Appendix A for an overlay of this plot with the traditionally allowed regions of various secondary structures in the Ramachandran map.

**Table 1 molecules-26-01533-t001:** Description of donor and acceptor orbitals of i+1-i+2 overlap interactions found in a β-turn.

Interaction	Donor (i+1)		Acceptor (i+2)		Covalent Bond
	Atom(s)	Orbital	Atom(s)	Orbital	Separation
$n\to \pi^{*}$	O	*n*	C=O	$\pi^{*}$	4
$n\to \sigma_{CC}^{*}$	O	*n*	$C_{\alpha }$-$C_{\beta }$	$\sigma^{*}$	3
$n\to \sigma_{CH}^{*}$	O	*n*	$C_{\alpha }$-$H_{\alpha }$	$\sigma^{*}$	3
$\pi \to \pi^{*}$	C=O	π	C=O	$\pi^{*}$	3

**Table 2 molecules-26-01533-t002:** Geometric parameters and stabilisation energies $E_{int}$ (kcal/mol) in the i+1−−i+2 pair of residues pertaining to the β-turn long-range interactions. Refer Figure 4 for definitions.

β-Variant	($\phi_{i+2}$,$\psi_{i+2}$)${}^{\circ }$	$n\to \pi^{*}$			$n\to \sigma_{\mathrm{CH}}^{*}$			$n\to \sigma_{\mathrm{CC}}^{*}$		$\pi \to \pi^{*}$		
		d(Å)	θ(${}^{\circ }$)	$E_{\mathrm{int}}$	Ω(Å)	ξ(${}^{\circ }$)	$E_{\mathrm{int}}$	τ(${}^{\circ }$)	$E_{\mathrm{int}}$	λ(Å)	Δ(${}^{\circ }$)	$E_{\mathrm{int}}$
vT	(−58.8,−46.6)	2.9	97.6	0.46	2.6	51.2	-	163.6	0.54	3.3	13.2	0.27
vA	(−59.7,−42.2)	3.0	102.2	0.25	2.7	44.6	-	163.6	0.51	3.5	15.4	0.18
vS	(−66.3,−28.9)	3.0	112.2	0.20	2.5	43.1	0.12	162.2	0.52	3.7	21.2	0.11
vV	(−85.1,−12.4)	3.3	122.8	-	2.4	24.5	0.89	150.1	0.41	4.2	15.8	-

## Data Availability

Data is contained within the article or Appendix A.

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
