# Peer review of "Cross-Talk between Overlap Interactions in Biomolecules: A Case Study of the β-Turn Motif"

_molecules, 2021, doi:10.3390/molecules26061533_

Round 1

Reviewer 1 Report

In this study, Nagesh investigated the quantum mechanical (QM) orbital overlap based interactions in beta-turns of the model protein GB1. The work shows that upon sidechain substitution, these interactions display compensatory effects in electronic stabilization of the local beta-turn conformation. Similar analysis was extended to the other proteins structures containing beta-turns, revealing a consistent range of interaction strengths in the order of ~1 kcal/mol. The work is nicely performed and well organized. Here are some suggestions that should be considered by the author to improve the presentation of the work:

  1. To better understand the overall structure of the model protein and the surrounding environment of the beta-turns, please provide a figure showing the GB1 protein and highlighting the locations of the beta-turns.
  2. Line 78, “by substituting Lys10 at the i + 1 position of L1 by D-Valine(“v”) and Thr11 at the i + 2 position by various L-amino acids (Alanine, Serine, Valine and Threonine) to obtain ‘’vA”,“vS”,“vV” and “vT” variants respectively.” Why these amino acids were chosen for the substitution?
  3. Figure 2, which amino acid pair is used for the illustration here? What does the magenta/blue color stand for?
  4. Line 95 “A polarizable continuum model was used to implicitly account for the water solvent.” Please provide more details of the surrounding environment (the continuum model).
  5. Line 131 “…even when the dihedral angle x goes to zero or 180;” this statement is confusing, the angle does not seem to be zero or 180. Please clarity it.
  6. Section 2.3, what are the amino acid compositions of the 22 proteins? Line 156 “The list of chosen proteins includes all the four types of b-turns (I,I’,II,II’).” Please label or group the protein with different types of beta-turns.
  7. The way the data is presented in Figure 6 is not clear, especially when several shapes are overlapped on each other. I suggest the author to show every data point (for each protein) as a small pie graph (circle) on this 2D plot, with the size of the circle scaled according to the magnitude of the total interaction energy of each protein, and the four different energies constitute the pie circle. By doing this, the overlapping of the data could avoided.
  8. Line 67, “in beta-turns of several proteins structures…” should be “protein structures”. Please check through the text to make sure there is no grammatical errors or typos.

Reviewer 2 Report

The article entitled “Cross-talk between overlap Interactions in biomolecules: a case study of the β-turn motif” is focused on the identification and characterization of all noncovalent overlap interactions in the β-turn, a key secondary structural element in the folding of a polypeptide chain.

The authors analyzed the QM framework of natural orbitals, reporting the occurrence of several interactions, including n à σ* and  π à π*, further the n à π*, that provide a large energetic contribution. The β-turn analysis of GB1 evidenced that these interactions are sensitive to changes in the type of residue composing the motif, thus deducing that all these interactions contribute to conformation and stability of the β-turn. The occurrence of such interactions was also reported by the analysis of various structures reported in the PDB, indicating that all overlap interactions contribute to protein structure and energetics. Moreover, geometric criteria were proposed by the authors based on the extent of QM overlaps and interaction energies, exploitable for broad database searches. The manuscript is well written and provides a deep analysis of this important structural element, reporting interesting results on the identification and characterization of all noncovalent overlap interactions in this motif.

I will recommend the acceptance after minor revisions (listed below).

Minor revisions:

  1. Keywords, page 1. Pdb is too generic and seems not an appropriate keyword for the manuscript, please change with something more specific or remove it.
  2. Amino acid names does not require capital letters, change it throughout the manuscript.
  3. Results, section 2.1, page 2 lines 79-80. Please change “by various L-amino acids (Alanine, Serine, Valine and Threonine) to obtain ‘’vA”,“vS”,“vV” and “vT” variants respectively.” in ““by the L-amino acids alanine, serine, valine and threonine to obtain the ‘’vA”,“vS”,“vV” and “vT” variants, respectively.”
  4. Results, section 2.1, page 3 line 83. The acronym NBO has been already used in the introduction, the is no need to repeat it.
  5. Results, section 2.3, page 6 line 150. Change “proteins” in “structures”
  6. Results, section 2.3, page 6 line 161. The term “cases” is repeated twice in the same line, please remove/cage one of them.
  7. Figure 6 and Figure S1 (Supporting Information). In the current form the symbols of the interactions are highly overlapped in some areas of the plot, thus it is difficult to seem some of them. The authors should generate a clearer version of this figure. The information about the magnitude could be separated in a table. Furthermore, Figure S1 is not called in the manuscript.
  8. Methods, section 4, page 9 line 249. The acronym NBO has been already used in the introduction, the is no need to repeat it.
  9. Methods, section 4. Please report here the criteria used to select the PDB structures used for the analysis.
  10. Page 9, line 260. Conclusions are not included; thus this section could be removed.

Reviewer 3 Report

This manuscript describes a very interesting study of non-covalent interactions that are contributing to stability of beta-turn motif in proteins. Besides well-studied n-pi* interactions, author investigates several additional interactions which should be taken into account when discussing the energetics of weak interactions and protein conformation. The results of this research suggest several areas for future investigations. Overall, I found this manuscript to be a very interesting read and one that is certainly worthy of publication in Molecules.

Author Response

I thank the reviewer for reading the manuscript and for the kind words of encouragement.